# Altered Mitochondrial Opa1-Related Fusion in Mouse Promotes Endothelial Cell Dysfunction and Atherosclerosis

**DOI:** 10.3390/antiox11061078

**Published:** 2022-05-28

**Authors:** Ahmad Chehaitly, Anne-Laure Guihot, Coralyne Proux, Linda Grimaud, Jade Aurrière, Benoit Legouriellec, Jordan Rivron, Emilie Vessieres, Clément Tétaud, Antonio Zorzano, Vincent Procaccio, Françoise Joubaud, Pascal Reynier, Guy Lenaers, Laurent Loufrani, Daniel Henrion

**Affiliations:** 1MITOVASC Department, Team 2 (CarMe), ICAT SFR, University of Angers, 3 rue Roger Amsler, F-49500 Angers, France; ahmadchehaitly2017@outlook.fr (A.C.); anne-laure.guihot@univ-angers.fr (A.-L.G.); coralyne.proux@univ-angers.fr (C.P.); linda.grimaud@univ-angers.fr (L.G.); jade.aurriere@gmail.com (J.A.); benoit.legouriellec@hotmail.fr (B.L.); jordan.rivron@etud.univ-angers.fr (J.R.); emilie.vessieres@univ-angers.fr (E.V.); clement.tetaud@univ-angers.fr (C.T.); vincent.procaccio@univ-angers.fr (V.P.); pareynier@chu-angers.fr (P.R.); guy.lenaers@inserm.fr (G.L.); laurent.loufrani@inserm.fr (L.L.); 2Institut National de la Santé et de la Recherche Médicale (INSERM) U1083, 3 rue Roger Amsler, F-49500 Angers, France; 3Centre National de la Recherche Scientifique (CNRS) UMR 6015, 3 rue Roger Amsler, F-49500 Angers, France; 4Institute for Research in Biomedicine (IRB Barcelona), The Barcelona Institute of Science and Technology, Baldiri Reixac, 10–12, 08028 Barcelona, Spain; antonio.zorzano@irbbarcelona.org; 5Department of Biochemistry and Molecular Biomedicine, Faculty of Biologie, University of Barcelona, 08028 Barcelona, Spain; 6Centro de Investigación Biomédica en Red de Diabetes y Enfermedades Metabólicas Asociadas (CIBERDEM), Instituto de Salud Carlos III, C/ de Monforte de Lemos, 5, 28029 Madrid, Spain; 7University Hospital (CHU) of Angers, 4 rue Larrey, F-49933 Angers, France; frjoubaud@chu-angers.fr

**Keywords:** mitochondrial fusion, blood flow, shear stress, arteries, endothelial cell, atherosclerosis

## Abstract

Flow (shear stress)-mediated dilation (FMD) of resistance arteries is a rapid endothelial response involved in tissue perfusion. FMD is reduced early in cardiovascular diseases, generating a major risk factor for atherosclerosis. As alteration of mitochondrial fusion reduces endothelial cells’ (ECs) sprouting and angiogenesis, we investigated its role in ECs responses to flow. Opa1 silencing reduced ECs (HUVECs) migration and flow-mediated elongation. In isolated perfused resistance arteries, FMD was reduced in *Opa1*^+/−^ mice, a model of the human disease due to Opa1 haplo-insufficiency, and in mice with an EC specific Opa1 knock-out (EC-Opa1). Reducing mitochondrial oxidative stress restored FMD in EC-Opa1 mice. In isolated perfused kidneys from EC-Opa1 mice, flow induced a greater pressure, less ATP, and more H_2_O_2_ production, compared to control mice. Opa1 expression and mitochondrial length were reduced in ECs submitted in vitro to disturbed flow and in vivo in the atheroprone zone of the mouse aortic cross. Aortic lipid deposition was greater in *Ldlr*^−/-^-*Opa1*^+/-^ and in *Ldlr*^−/-^-EC-Opa1 mice than in control mice fed with a high-fat diet. In conclusion, we found that reduction in mitochondrial fusion in mouse ECs altered the dilator response to shear stress due to excessive superoxide production and induced greater atherosclerosis development.

## 1. Introduction

The vascular endothelium has a central role in vascular homeostasis, and an endothelial dysfunction is a hallmark of cardiovascular disorders [1]. Blood flow generates shear stress on the intraluminal side of the endothelial cells (ECs) and activates the production of vasodilator agents such as nitric oxide (NO) [2,3]. This flow-mediated dilation (FMD) is reduced early in patients with metabolic disorders, such as diabetes and obesity, and in patients suffering from cerebrovascular and cardiovascular diseases [4]. Decreased FMD is also a predictor of cognitive decline in neurological diseases [5]. A reduction in FMD and more widely in endothelium-dependent dilation, is mainly the result of excessive oxidative stress [6], in part of mitochondrial origin [6,7]. Indeed, ECs are sensitive to mitochondrial oxidative stress and ECs mitochondria undergo hyper-fragmentation in cardiovascular diseases [8]. Although ECs do not depend on mitochondria for their energy needs, as they mainly rely on glycolysis to produce ATP [9], it is hypothesized that mitochondria play a regulatory role in ECs [10].

Mitochondria are highly dynamic structures, changing their inner and outer membranes through fission and fusion mechanisms. The fission process is mediated by the cytosolic dynamin proteins DRP1 and DNM2, and the outer-membrane human fission factors Fis1, MFF, MID49, and MID51. On the other hand, mitochondrial fusion is mediated by the outer-membrane Mitofusins 1 and 2 (MFN1, 2) and the inner-membrane optic atrophy factor 1 (OPA1) [11]. Pharmacological inhibition of mitochondrial fission has protective activity in heart ischemia-reperfusion injury [12]. In pulmonary artery smooth muscle cells hyper-proliferation is associated with mitochondrial fragmentation in hypertensive mice [13]. Similarly, mitochondrial fission in smooth muscle cells is essential for PDGF-induced proliferation [14]. In contrast with smooth muscle cells, the pathway regulated by mitochondrial dynamics in ECs is less described. Nevertheless, fatty acid and amino acid oxidation pathways are involved in angiogenic sprouting [15]. Furthermore, mitochondrial fusion and fission are sensitive to shear stress in cultured ECs [16] and shear stress-dependent Ca^2+^ mobilization in ECs relies on mitochondria-dependent activation of endoplasmic reticulum channels [17]. A recent study has demonstrated that ECs’ OPA1 is required in developmental and tumor angiogenesis [18]. This work has also shown that OPA1 is required for ECs’ migration, branching, and filopodia formation in tip ECs during angiogenesis [18]. Thus, we hypothesized that mitochondrial fusion could play a role in the endothelium-dependent protection of the vascular wall through its involvement in flow-mediated signaling [3,19]. To address this question, we studied mouse models with *Opa1* knock-out in endothelial cells and with *Opa1* systemic haplo-insufficiency (*Opa1^+^*^/*−*^). The *Opa1^+^*^/*−*^ mouse model reproduces the autosomal dominant optic atrophy (ADOA) syndrome in humans [20]. Although ophthalmic and neurologic disorders are the main ADOA phenotypic characteristics in humans [21], vascular dysfunction could occur as suggested by the observation of blood vessels tortuosity [22] and decreased density in the retinal circulation in ADOA patients [23]. These vascular defects might reflect an endothelial dysfunction [24]. Moreover, Opa1 downregulation due to diabetes takes part in the development of retinal vascular lesions [25]. The occurrence of metabolic stroke has also been observed in a patient with “ADOA*plus* syndrome” [26]. Finally, we have recently shown that the *Opa1*^+/−^ mouse is more susceptible to hypertension [27]. Thus, we investigated the role of mitochondrial fusion in FMD and cardiovascular disease development, with the perspective to position mitochondrial dynamics as a tractable therapeutic target [11].

## 2. Materials and Methods

### 2.1. Mice

Heterozygous *Opa1^+^*^/*−*^ male and female mice carrying the recurrent *Opa1*-c.2708*^delTTAG^* mutation were used as a model of reduced mitochondrial fusion reproducing the ADOA disease [20].

Mice lacking Opa1 in endothelial cells were generated after crossing *Cadherin5-CreERT2* mice with *Opa1^loxP^*^/*loxP*^ mice [28]. They are designed as EC-Opa1 mice (*Cadherin5-CreERT2^+^Opa1^loxP^*^/*loxP*^) compared to their littermate control EC-WT mice (*Cadherin5-CreERT2^−^Opa1^loxP^*^/*loxP*^). The deletion is induced by injection of tamoxifen (150 mg/kg per day, diluted in corn oil) for 5 consecutive days. Mice were used at least 5 weeks after tamoxifen induction.

To generate mice susceptible to atherosclerosis, *LDLr**^−^*^/*−*^ female mice [29] were crossed with male *Opa1^+^*^/*−*^ or male EC-Opa1 mice.

Mice obtained and used for the present study were:− *Ldlr^−^*^/*−*^*Opa1*^+/−^ and their littermate control *Ldlr^−^*^/*−*^*Opa1*^+/+^− *Ldlr*^−/−^EC-Opa1 mice and their littermate control *Ldlr*^−/−^EC-WT.

In all protocols, littermate wild-type mice were used as control. Each mouse was genotyped before entering a protocol. All procedures were performed in accordance with the principles and guidelines established by the National Institute of Medical Research (INSERM) and were approved by the local Animal Care and Use Committee (APAFIS#3577-201601141342229, APAFIS#2018011217209, APAFIS#30385-2021031010145750). The investigation conforms to the directive 2010/63/EU of the European parliament.

### 2.2. Flow-Mediated Dilation in Mesenteric Arteries In Vitro

A segment of mesenteric resistance artery, approximately 250 µm in luminal diameter, was cannulated on small glass micro-pipettes connected to a video-monitored perfusion system (Living System, LSI, Burlington, VT, USA) [30]. The arterial segment was placed in a 5 mL organ bath containing a physiological salt solution (PSS) of the following composition (mmol/L): 130.0, NaCl; 15.0, NaHCO_3_; 3.7, KCl; 1.6, CaCl_2_; 1.2, MgSO_4_ and 11.0, glucose. pH was 7.4, pO_2_ 160 mmHg and pCO_2_ 37 mmHg). The arterial segment was perfused using a first peristaltic pump controlling the flow rate and a second peristaltic pump controlling the perfusion pressure thanks to a pressure-servo control unit. Pressure was set at 75 mmHg. Arterial contractility was tested with KCl (80 mmol/L) and phenylephrine 1 µmol/L. Endothelium-dependent dilation was tested using acetylcholine (1 µmol/L, with a precontraction of 50%). Flow (3 to 50 µL per min) was then generated to determine flow-mediated dilation (FMD) [31]. In some experiments, FMD was determined in the presence of N(omega)-nitro-L-arginine (L-NNA, 10^−4^ mol/L, 30 min) or SOD (120 U/mL, 30 min) plus catalase (80 U/mL, 30 min) [32], or MitoTempo (1 µmol/L, 30 min) [33], or tetrahydrobiopterin (BH4, 10 µmol/L, 30 min) plus L-arginine (L-Arg, 100 µmol/L, 30 min) [32] or the Opa1 blocker MYSL22 (1 µmol/L, 30 min) [18].

At the end of each protocol, the arterial segment was bathed in a Ca^2+^-free PSS containing ethylene-bis-(oxyethylenenitrolo) tetra-acetic acid (2 mmol/L) and sodium nitroprusside (10 µmol/L) before increasing pressure by step from 10 to 125 mmHg (no flow) in order to determine arterial diameter and wall thickness [32].

As these experiments involved living arteries some experiments were halted before the end of a protocol. Thus, 10 to 12 mice were used per group depending on litter size and on the number of KO and WT mice per litter. Because of tissue viability, only 2 or 3 arterial segments could be collected per mouse. The main reasons for stopping an experiment were: (1) breaking of a glass cannula, breaks during the mounting of the artery or during a washout, (2) perfusion pressure could not be maintained due to a leak (small collateral arteriole not visible when mounting the artery), (3) the artery does not pass a test: no contractility (smooth muscle damaged during dissection or mounting) or no dilation (endothelium damaged during dissection or mounting), (4) disconnection of the artery from one glass cannula before the end of the experiment, (5) formation of gas bubbles in the tubing and one bubble crosses the artery. This is enough to damage the endothelium and to stop the experiment. Consequently, the number of mice used per group was not homogenous. Number of mice included is given in the figure legends.

### 2.3. Perfused Isolated Mouse Kidney

In another group of experiments the right renal artery was cannulated in anesthetized mice. The right kidney was then excised and perfused as described previously [34]. Perfusion rate was 600 µL/min and the perfusion pressure was monitored using a PM-4 pressure monitor (LSI, Burlington, VT, USA). Endothelium-dependent dilation was assessed through the perfusion of acetylcholine (1 µmol/L) after precontraction (phenylephrine, 1 µmol/L). The perfusion flow was then increased by step in order to determine the flow-pressure relationship [34].

The PSS perfusing the kidney was collected when the flow was 600 µL/min. The perfusate was immediately frozen in liquid N_2_ and then stored at −80 °C.

As these experiments involved living kidneys some experiments were halted before the end of a protocol. Thus, 8 mice were used per group and one kidney could be used per mouse. The main reasons for stopping an experiment were: (1) gas bubble moving from the tubing to the kidney, (2) no contractility or no dilation if the dissection or cannulation was too long, (3) disconnection of the kidney from the cannula. Consequently, the number of mice included per group was not homogenous. The number of mice included is given in the figure legend.

### 2.4. Determination of ATP and H_2_O_2_ Levels in the Kidney Perfusate

Flow-induced ATP and H_2_O_2_ release were measured on perfusate collected from each kidney (500 µL) and concentrated as previously described [34]. Hydrogen peroxide (H_2_O_2_) concentration was then measured using the Hydrogen Peroxide Assay Kit Colorimetric/Fluorometric NbAb102500, Abcam, Cambridge, UK) and the concentration of ATP was determined using the ATP Determination Kit (Nb A22066, Invitrogen^TM^ Molecular Probes, Eugene, OR, USA) [34].

### 2.5. Cell Culture

Human Umbilical Vein Endothelial Cells (HUVECs) were obtained from Lonza (Basle, Switzerland) and cultivated in endothelial cell growth medium-2 (EGM-2, Lonza, Basle, Switzerland) supplemented with the EGM^TM^-2 SingleQuots kit (Lonza, Basle, Switzerland), at 37 °C and 5% CO_2_. Cells from passages 1 to 5 were used for experiments. For orbital shaker experiments, the MS1 (MILE SVEN1) cell line was used (endothelial C57BL6 cell line obtained from the islets of Langerhans, ATCC^®^CRL2279^TM^, Les Ulis, France). Cells were maintained in a DMEM culture medium (Lonza, Basle, Switzerland) supplemented with 5% fetal bovine serum (Eurobio scientific, Les Ulis, France), glutamine (2 mM, Lonza, Basle, Switzerland), penicillin (100 U/mL, Lonza, Basle, Switzerland), and streptomycin (100 µg/mL, Lonza, Basle, Switzerland). Cells were maintained at 37 °C with 5% CO_2_ and 95% humidity.

### 2.6. RNA Interference

RNA interference experiments were performed by transfection of a silencer scrambled negative control (Ambion, cat# AM4611, 50 nM, Warrington, UK) and pre-validated silencer selected pre-designed siRNAs against OPA1 (Ambion cat# s9851, 50 nM, Warrington, UK) using the Lipofectamine™ transfection reagent (Invitrogen cat# 18324012, Eugene, OR, USA), according to the manufacturer’s instructions. HUVECs were seeded at a density of 750,000 cells/well in a 6-well plate. The lipofectamine and the siRNA were diluted in a reduced serum Opti-MEM (Gibco, Grand Island, NY, USA), mixed in a 1:1 *v*/*v*, and incubated for 30 min at room temperature. Then, 250 μL of the mixture was added to each well of the 6-well plate with 500 μL of Endothelial Cell Growth Basal Medium-2 (EBM^TM^-2) from Lonza (Basle, Switzerland), for 48 h. To verify siRNA efficiency, a western blot assay of Opa1 was carried out. Transfected cells were trypsinized, washed three times in ice-cold PBS, and harvested using SDS lysis buffer (SDS 1% *v*/*v*, Tris 10 mM pH7.4, halt protease and phosphatase inhibitor cocktail, 5 mM EDTA solution).

### 2.7. Cell Migration Assay

For the migration assay, Ibidi^®^ culture inserts 2-well in μ-well (cat# 81176, Ibidi, Munich, Germany) were used. Briefly, the inserts consist of two chambers separated by a 0.5 mm divider, each chamber with a growth area of 0.22 cm^2^. The inserts were planted into 35 mm tissue culture dishes using sterile tweezers. HUVECs were transfected using the previously described protocol, cell suspensions were prepared at 7–9 × 105 cells/mL in medium, of which 70 μL was transferred to each chamber, allowing cell adhesion and growth to ~100% confluence. After removing the strip between dividing the chambers image acquisition was performed along the cell-free zone from 0 to 8 h. Cell migration was quantified using the ImageJ—Fiji software.

### 2.8. In Vitro Exposure of Endothelial Cells to Shear Stress

The Ibidi^®^ pump system (Fluidic Unit: Cat No. 10902; Perfusion set 15 cm, ID 1.6 mm: Cat No. 10962; μ-slide I 0.4 Luer: Cat No. 80176; Ibidi GmbH, Munich, Germany) was used to submit ECs to flow. The pump system was constructed according to the instructions (Ibidi GmbH, AN 13: HUVECs under Perfusion, Munich, Germany). 350,000 primary HUVEC cells were seeded onto a μ-slide I 0.4 Luer and incubated for 24 h at 37 °C and 5% CO_2_ to form a monolayer. The next day, the experiment was conducted under the following conditions: 31.9 mbar pressure, 21.70 mL/min flow rate, 0.007 dyn x s/cm^2^ viscosity and a shear stress of 20 dyn/cm^2^. Static conditions were also tested on the same Ibidi^®^ μ-slide I 0.4 Luer (Ibidi, Munich, Germany).

### 2.9. Measurement of Endothelial Cells Alignment and Elongation

Endothelial cells’ alignment was assessed using the Directionality plugins in Image J—Fiji (https://imagej.net/Directionality, available online https://imagej.net/software/fiji/ accessed on 28 June 2012). Quantification of cell orientation, measured by elongation factor (cell length along flow direction divided by cell width) [35] was performed on the same images.

### 2.10. Circular Flow Assay

In order to submit a larger number of cells to flow, orbital shaker experiments were performed using the MS1 (MILE SVEN1) cell line (ATCC^®^CRL2279TM, Les Ullis, France). Cells were maintained in a DMEM culture medium (Lonza, Basle, Switzerland) and supplemented with 5% *v*/*v* fetal bovine serum, 2 mM glutamine, 100 U/mL penicillin and 100 µg/mL streptomycin, in a humid atmosphere of 37 °C with 5% CO_2_. Cells were then placed at 37 °C as control and cells under flow were placed under agitation of our laboratory digital orbital shaker for 24 or 72 h at 210 rpm (12 dyn/cm^2^) or 260 rpm [36].

In order to verify that cells were submitted to laminar (protective) flow we measured *Nos3* (encoding for eNOS) and *Edn1* (encoding for ET-1) gene expression. Laminar flow has been shown to increase eNOS epression and to decrease *Edn1* expression whereas a disturbed flow has the opposite effect [37,38,39,40,41].

Cells were then collected for transcriptomic analyses as described below.

### 2.11. Analysis of mRNA Levels by RT-qPCR

We used quantitative polymerase chain reaction after reverse transcription of total RNA (RT-qPCR) in order to measure gene expression in the mouse aorta. Segments of the aorta were stored at −20 °C in RNAlater Stabilization Reagent (Qiagen, Valencia, CA, USA). In each segment of aorta, the RNA was extracted (RNeasy^®^ Micro Kit, Qiagen, Valencia, CA, USA) according to the instructions provided by the manufacturer. 500 ng) of the RNA extracted from the aortic segments was then used to synthesize cDNA using the QuantiTect^®^ Reverse Transcription Kit (Qiagen, Valencia, CA, USA). RT-qPCR was thenperformed using the Sybr^®^ Select Master Mix (Applied Biosystems Inc., Lincoln, CA, USA) in a LightCycler 480 Real-Time PCR apparatus (Roche, Branchburg, NJ, USA).

Primers were validated by testing PCR efficiency using standard curve as MIQE guidelines [42]. Mouse Primers (Eurogentec, Liège, belgium) were presented in Table 1, *Hprt, Gapdh* and *Gusb* were used as housekeeping genes.

Analysis was not performed when Cq values exceeded 35. Gene expression was quantified using the comparative Cq method. Expression values shown were normalized per mRNA as fold changes of means of WT mice.

### 2.12. Immunofluorescence Analyses

For immunofluorescence staining, the thoracic aorta was opened longitudinally. The greater and the lower curvature of the aortic cross were dissected and then fixed for 10 min with 4% *w*/*v* paraformaldehyde and permeabilized for 10 min with 0.1% *v*/*v* Triton X-100 in PBS with 1% *w*/*v* BSA, followed by 50 min blocking with 1% *w*/*v* BSA before probing with various antibodies. Nuclei were visualized with DAPI (Merck Cat#D9542, 25 µg/mL PBS with 1% *w*/*v* BSA, Laser intensity was 50%; 500 ms exposure, Darmstadt, Germany). The endothelium was visualized using VE-cadherin antibody, Rat (cat#14-1441-82, Invitrogen, 1/100, Eugene, OR, USA) coupled with goat anti-rat IgG (H + L) highly cross-adsorbed secondary antibody, Alexa Fluor 488 (cat#A11006, Invitrogen, 1/500, laser intensity was 50%; 1000 ms exposure time, Eugene, OR, USA). Mitochondria were visualized using TOMM20 recombinant rabbit monoclonal antibody (Invitrogen™, cat#MA5-32148, 1/200, Eugene, OR, USA) coupled with donkey anti-rabbit IgG Alexa Fluor 568, Invitrogen™ (cat#A10042, Invitrogen, 1/100, laser intensity was 50%, 2000 ms exposure, Eugene, OR, USA). Staining of negative controls was performed under the same conditions with, respectively, Mouse IgG1 kappa isotype control (P3.6.2.8.1), Alexa Fluor 488, (cat#53-4714-80, eBioscience™, Waltham, MA, USA) and anti-rabbit IgG Alexa Fluor 568. Staining was visualized using confocal microscopy and Metamorph software (Molecular devices). Image analysis was performed using ImageJ—Fiji.

### 2.13. Mitochondrial Shape Measurement in Endothelial Cells

The mitochondrial shape was analyzed using the Image J—Fiji software (version: 2.1.0/1.53c) as previously described [43,44] on the endothelial cells stained as described above. Images were acquired using Metamorph software and imported into ImageJ. Images were then processed using the MINA (mitochondria network analysis) plug in, providing mitochondria area and number [45]. The mitochondrial fission count (MFC) was then measured as it represents the mitochondria dynamics. MFC was calculated as the total number of mitochondria in a cell divided by the mitochondrial area in the same cell [44].

### 2.14. Analysis of Protein Expression Levels by Western Blot

HUVECs submitted to flow were extracted in RIPA lysis buffer. Homogenates were centrifuged at 13000 rpm at 4 °C for 20 min, and the resulting supernatant was collected. Protein concentration was determined using the Micro BCA protein assay kit (cat#23227, Thermo Fisher Scientific, Waltham, MA, USA) according to the manufacturer’s instructions. Equal amounts of proteins (10 µg) were solubilized under denaturating conditions in 25 µL of Laemmli sample buffer containing 2.5% *v*/*v* β-mercaptoethanol, boiled 5 min at 90 °C, separated by 4–15% polyacrylamide gel electrophoresis (BioRad, Marnes la Coquette, France) and transferred to a nitrocellulose membrane (BioRad). Membranes were incubated overnight at 4 °C with the primary antibody followed by the appropriate peroxidase-labeled secondary antibody (Table 2) for 1 h. Reactions were visualized by ECL detection according to the manufacturer’s instructions (Bio-Rad, Marnes-la-Coquette, France) and membranes were stripped at 50 °C for 30 min in the presence of beta-mercaptoethanol before re-blotting.

### 2.15. Endothelial Cell Isolation from Mouse Mesenteric Arteries

In order to validate the extinction of Opa1 in EC-Opa1 mice, we isolated endothelial cells from mesenteric arteries. A total of two mice were used to obtain enough endothelial cells for one experiment. Mice were anesthetized and exsanguinated. The mesenteric bed was then ligated below the duodenum and above the caecum and excised. All the arteries from the first to the fourth order of the mesenteric bed were dissected. They were placed in a digestion solution composed of Endothelial Cell Growth Basal medium 2 (Lonza, Basle, Switzerland) supplemented with 1000 IU type 2 collagenase. The arteries were cut into small segments using sterilized micro-scissors and left to stir in a 37 °C and 5% CO_2_ incubator for 45 min. The homogenized suspension obtained was filtered through a 30 μm Miltenyi filter and then centrifugated (10 min, 300 g). Depending on the number of cells counted, a sufficient quantity of CD146-labeled Miltenyi Microbeads was added to the cell pellet, and left to incubate for 15 min at 4 °C. The cells were then washed before performing a magnetic separation on an MS Miltenyi column. Cells not retained by the column were seeded after washout in DMEM F12 10% *v*/*v* FCS in a well of a 6-well plate. They were identified as the non-endothelial cells. Cells retained by the column were flushed and then washed before being seeded into a well of a 12-well plate in Endothelial Growth medium 20% *v*/*v* FCS. After 24 to 48 h of adhesion, the cells were washed and allowed to grow until confluence in EGM-2 medium (Lonza, Basle, Switzerland).

### 2.16. Analyses of Atherosclerosis Lesions

Male and female *Ldlr^−^*^/*−*^*-Opa*^+/−^, *Ldlr^−^*^/*−*^*-Opa*^+/+^, *Ldlr*^−/−^-EC-Opa1 and *Ldlr*^−/−^-EC-WT mice aged ten-weeks old were subjected to a hypercholesterolemic atherogenic diet (1.25% w/w cholesterol, 7.7% w/w fat, no cholate, U8959P version 0022, SAFE, Lyon, France). After 4 months, lipid deposition size was estimated as previously described [46]. Briefly, the abdominal cavity was opened, and the internal organs were removed. The aortic sinus and the aorta from the arch to the iliac bifurcation were carefully cleaned of peri-adventitial connective tissue. Then the aorta was opened longitudinally by an incision along its ventral aspect. The thoracic and abdominal parts of the aorta were pinned out flat (using 10-mm tips of dental root-canal needles). The aorta and the sinus were stained with oil red O (cat#O0625, Merck, Darmastadt, Germany) and counterstained with Mayer’s hematoxylin. The percentage of the areas with lesions was evaluated in each sample [46].

### 2.17. Lipids and Glucose Blood Level in Mice

The concentrations of glucose, total cholesterol, HDL cholesterol, LDL cholesterol, and triglycerides in the sera collected from the mice were assayed on ADVIA ChemistryXPT (Siemens) system, using Liquid Assayed Multiqual control sera, according to the manufacturer’s recommendations.

### 2.18. Statistical Analyses

For FMD and the measurements of pressure-dependent increases in diameter and wall thickness, a 2-way ANOVA for repeated measurements (flow or pressure steps) was performed. When a significant interaction was observed between 2 factors, the effect of the gene deletion was studied using a Bonferroni post-test.

In the other experiments, the two-tailed Mann–Whitney test was used when 2 groups were compared and the Kruskal–Wallis test was used when more than 2 groups were involved. The test used is indicated in each figure legend. A probability value lower than 0.05 was considered significant.

## 3. Results

### 3.1. In Vitro Endothelial Cell Migration

OPA1 expression was reduced by 76% in HUVECs treated with the siOPA1 compared to control cells treated with the scrambled siRNA (siCONT) (Appendix A). The siOPA1 induced a reduction in endothelial cell migration, compared to the scrambled siRNA (Figure 1A–E).

A positive and a negative control were performed with VEGF and a high concentration of glucose in the medium, respectively (Appendix A).

### 3.2. In Vitro Flow-Mediated Endothelial Cell Alignment and Elongation

HUVECs subjected to laminar shear stress were similarly aligned in the direction of flow after treatment with siOPA1 compared to cells treated with scrambled siCONT (Figure 1F–J). Nevertheless, the elongation factor (ratio of cell length/cell width) was lower in siOPA1-treated cells (OPA1 silencing = 89%, Appendix A) compared to ECs treated with scrambled siCONT (Figure 1K–M).

### 3.3. OPA1 Silencing Altered the Response of Endothelial Cells to Laminar Flow

OPA1 silencing in HUVECs submitted to laminar flow induced a 78% reduction in OPA1 expression (Figure 2A), without affecting eNOS level (Figure 2B). Nevertheless, OPA1 silencing reduced Klf2, PFKFB3, SOD2, and Cu/Zn SOD expression levels (Figure 2C–E), while increasing p22phox level, without affecting gp91 expression (Figure 2G,H). These results suggest that the absence of OPA1 increased oxidative stress in ECs.

### 3.4. In Vitro Flow-Mediated Dilation in Opa1^+/−^ Mouse Resistance Arteries

The occurrence of Opa1 haploinsufficiency in mesenteric arteries isolated from *Opa1^+^*^/*−*^ mice was demonstrated in our previous work showing a 69% Opa1 reduction in mesenteric resistance arteries [27]. Stepwise increases in flow rate in mesenteric resistance arteries induced a progressive dilation or FMD (Figure 3A). FMD was significantly reduced in resistance arteries isolated from male *Opa1*^+/−^ mice, compared to littermate *Opa1*^+/+^ male controls (Figure 3A). Arterial diameter (Figure 3B) and wall thickness (Figure 3C) were equivalent in male *Opa1*^+/−^ and *Opa1*^+/+^ mice. Acetylcholine-dependent dilation was not significantly different between *Opa1*^+/−^ and *Opa1*^+/+^ male mice (Figure 3D). A similar pattern was observed in female mice (Figure 3E–H).

### 3.5. Flow-Mediated Dilation in Mouse Lacking Opa1 in Endothelial Cells

Flow-mediated dilation was then measured in isolated mesenteric resistance arteries isolated from mice lacking Opa1 in endothelial cells (EC-Opa1) compared to controls (EC-WT) (Figure 4A). Flow-mediated dilation was significantly reduced in male EC-Opa1 mice compared to EC-WT, while arterial diameter (Figure 4B) and wall thickness (Figure 4C) were not affected by Opa1 absence in EC-Opa1. Acetylcholine- and SNP-dependent dilation (Figure 4D,E) were not significantly different between EC-Opa1 and EC-WT mice. Similarly, contraction induced by phenylephrine (1 µmol/L, Figure 4F) or KCl (80 mmol/L, Figure 4E) was not affected by the absence of Opa1 in ECs. A similar pattern was observed in female mice (Figure 4H–N).

Blockade of NO synthesis with L-NNA significantly reduced FMD in both male and female EC-Opa1 and EC-WT mice (Figure 4A,H).

Blockade of Opa1 with the Opa1 inhibitor MYSL22 significantly reduced FMD in WT mice (Figure 4O).

Opa1 silencing was 61% in endothelial cells from resistance mesenteric arteries isolated from EC-Opa1 mice. No change in Opa1 level was observed in smooth muscle cells isolated from the same vessels (Figure 4P). Whole blots are shown in Appendix A.

Superoxide reduction with SOD and catalase restored FMD in male and female EC-Opa1 mice so that no difference between EC-Opa1 and EC-WT was observed after incubation of the arteries with SOD and catalase (Figure 5A,B). Similarly, after blocking superoxide production by the mitochondria with MitoTempo, no difference in FMD between EC-Opa1 and EC-WT mice was observed (Figure 5C,D).

FMD was also measured after the addition of BH4 and L-Arg. In this condition, FMD was equivalent in EC-Opa1 and EC-WT mice (Figure 5E,F).

### 3.6. Flow-Pressure Relationship, ATP and H_2_O_2_ Production in Ex-Vivo Perfused Kidney

As the kidney is a well autoregulated organ with a dense microvascular network, we investigated flow-mediated responsiveness in perfused kidneys isolated from EC-Opa1 and EC-WT mice (Figure 6A). First, the flow-pressure relationship was shifted upward in EC-Opa1 compared to EC-WT male and female mice (Figure 6B,C), suggesting that endothelial responsiveness to flow was reduced, in agreement with the reduced FMD observed in mesenteric arteries. Then, we measured ATP and H_2_O_2_ production in the kidney perfusate. Interestingly, ATP production measured in the kidney perfusate was reduced (Figure 6D,E) whereas H_2_O_2_ production was higher (Figure 6F,G) in EC-Opa1 compared to EC-WT male and female mice.

Mice body weight (Figure 6H), kidney/body weight (Figure 6I), acetylcholine-mediated dilation (Figure 6J) and phenylephrine-mediated contraction (Figure 6K) were similar in the perfused kidney from male and female EC-Opa1 mice compared to WT-Opa1 mice.

### 3.7. Disturbed Flow In Vivo and In Vitro Reduced Opa1 Level and Mitochondrial Length

In the mouse aorta, *Opa1* gene expression was higher in the greater curvature (GC) than in the lower curvature (LC) (Figure 7A). On the opposite, *Dnm1l* gene expression was lower in the GC than in the LC of the aortic cross (Figure 7B). This was observed in mice receiving a normal diet (Figure 7A,B) and in mice receiving a Western diet (Figure 7C,D).

The mitochondrial network density in the two aortic zones (LC in Figure 7E and GC in Figure 7F) was assessed by the measurement of the number of mitochondria (Figure 7G) and by the percentage of mitochondria on the cell surface (Figure 7H). These two parameters were similar in the LC and in the GC of the aortic cross. Nevertheless, mitochondrial fission count was lower (Figure 7I) and branch length was greater in the GC compared to the LC (Figure 7J), suggesting an alteration in mitochondrial dynamics in the LC.

As the LC of the mouse aortic cross is submitted in vivo to disturbed blood flow, we subjected mouse endothelial cells (MS1) in vitro, to a laminar or to a disturbed flow using an orbital shaker, submitting cells to a circular flow. Laminar flow was associated with an elevation in eNOS (*Nos3*) gene expression (Figure 7K) and a decrease in endothelin-1 (*Edn1*) gene expression (Figure 7L). In these conditions, *Opa1* and *Dnm1l* expression levels were not modified by laminar flow (Figure 7M,N, respectively). On the other hand, under conditions of disturbed flow, evidenced by a decrease in *Nos3* gene expression (Figure 7O) and an increase in *Edn1* gene expression (Figure 7P), *Opa1* expression level was decreased (Figure 7Q) while *Dnm1l* expression level was increased (Figure 7R).

### 3.8. Lipid Deposits in the Aorta of Opa1^+/−^ Mice

After 4 months of high-fat diet, mice had lipid deposits in the aorta (Figure 8). Lipid deposit was greater in male *Opa1*^+/−^ mice compared to male *Opa1*^+/+^ mice (Figure 8A) without significant differences in mouse body weight (Figure 8B), total cholesterol (Figure 8C), triglycerides (Figure 8D) and glycemia (Figure 8E) between the two groups. In mice submitted to a standard diet, no lipid deposit could be observed (Pictures on the left).

### 3.9. Lipid Deposits in the Aorta of Mice Lacking Opa1 in Endothelial Cells

Lipid deposition was greater in male EC-Opa1 mice compared to male EC-WT mice (Figure 9F). Bodyweight (Figure 9G), total cholesterol (Figure 9H), triglycerides (Figure 9I), and glycemia (Figure 9J) were similar in male EC-Opa1 and EC-WT mice. In mice submitted to the standard diet, no lipid deposit could be observed (Pictures on the left).

## 4. Discussion

The present study demonstrates that the mitochondrial fusion protein Opa1 is involved in ECs’ response to acute changes in flow (FMD), with a protective effect against atherosclerosis. This effect was equivalent in male and female mice.

Blood flow induces shear stress at the surface of ECs leading to the activation of protective pathways [4]. Laminar shear stress is a stabilizing stimulus for ECs leading to a nonproliferative, nonmigratory and antithrombotic phenotype [47,48]. An acute increase in flow induces a rapid vasodilation (FMD) in vitro and in vivo [31,34,49,50,51]. FMD measurement in humans has revealed a close relationship between the occurrence of risk factors and endothelial dysfunction [4].

Although ECs contain very few mitochondria [52] and do not depend on mitochondrial function for energy production [10,15,53], recent studies have shown that angiogenic factors such as VEGF activate endothelial glycolytic metabolism in ECs and that fatty acid and amino acid oxidation is involved in angiogenic sprouting [15]. Moreover, a recent study has shown that the mitochondrial fusion protein Opa1 limits NFkβ signaling in response to angiogenic stimuli, thus promoting angiogenesis [18]. This latter study highlights the role of Opa1 in angiogenesis and uncovered a major role of ECs’ mitochondria, independent of energy production. It further shows that Opa1 silencing in HUVECs and extinction in mouse ECs does not affect ECs’ ATP content, respiration and mitochondrial membrane potential. In this respect, our results showing a reduction in ECs migration and flow-mediated elongation in ECs submitted to Opa1 silencing, agree with these observations.

Mitochondrial fusion and fission are sensitive to flow (shear stress) in ECs [16] and flow-dependent Ca^2+^ mobilization relies on mitochondria-dependent activation of endoplasmic reticulum channels [17]. We observed that FMD was reduced in resistance arteries isolated from *Opa1^+^*^/*−*^ mice, the mouse model of ADOA, and from mice lacking Opa1 in ECs. These findings suggest a selective defect in the response of ECs to laminar shear stress. FMD is a rapid dilatory response of blood vessels depending on the ability of ECs to produce vasodilator agents such as nitric oxide (NO) [54]. Interestingly, FMD was also reduced in vitro by the specific Opa1 inhibitor MYLS22 [18] after a short (30 min) period of incubation, suggesting that Opa1 is involved dynamically in FMD rather than through changes in gene expression of the actors of flow-mediated mechanotransduction. It is most likely that Opa1 is involved selectively in the endothelial response to flow as the dilation induced by acetylcholine was not affected by Opa1 haploinsufficiency or by the absence of Opa1 in ECs. Similarly, SNP-dependent dilation which is independent of the endothelium and arterial contractility were not affected. Thus, the deficiency in endothelial Opa1 seems to affect mainly the endothelial response to flow.

We found that FMD was mainly dependent on the production of NO as L-NNA reduced FMD greatly. This observation agrees with previous studies in mouse and rat mesenteric arteries [31,55,56] as well as human coronary arteries in vitro [7] and in vivo as assessed at the level of the brachial artery [50]. Although FMD was reduced in mice lacking Opa1 in ECs, NO synthesis blockade with L-NNA remained able to further reduce FMD in these mice. In parallel, we found that the endothelium remained fully efficient in response to acetylcholine, which also activates NO production [57]. Acetylcholine and flow activate the production of NO by endothelial cells through different pathways. Acetylcholine activates eNOS phosphorylation of Ser1177 through the activation of Ca^2+^/calmodulin-dependent protein kinase II whereas flow acts through the activation of Akt [57]. This selective reduction in FMD without change in receptor-dependent dilation suggests an involvement of Opa1 in the pathway activated by flow (shear stress) only.

The decrease in FMD observed in mice lacking Opa1 in ECs could be the consequence of an excessive ROS production, at least in part of mitochondrial origin, as superoxide reduction by the combination of SOD and catalase improved FMD in these arteries. In agreement, we found that Opa1 silencing in ECs submitted to laminar flow reduced SOD expression and increased p22phox expression, suggesting that Opa1 is involved in the reduction in oxidative stress in ECs submitted to laminar flow. This observation is further confirmed by our results showing that in mice lacking Opa1 in ECs, the kidney produced less ATP and more H_2_O_2_ in response to flow. ATP is released by ECs in response to flow to activate purinergic receptor-dependent NO production [58,59]. On the other hand, H_2_O_2_ results from excessive oxidative stress as shown in arteries of patients suffering from coronary artery disease, which produce superoxide and H_2_O_2_, instead of NO in response to flow [7]. Other studies suggested that mitochondrial dynamics modulate ECs functions in pathological conditions, favoring superoxide production [60,61,62]. Thus, Opa1 is likely to facilitate the response of ECs to laminar flow through a reduction in oxidative stress, thus allowing a better NO bioavailability and consequently a better FMD. Indeed, flow (shear stress) simultaneously activates the production of NO and a reduction in the production of ROS and inflammatory agents [19]. The present study suggests that Opa1 is rather involved in the second arm of the response to flow, thus leading to flow-mediated reduction in ROS production. The resulting effect on NO bioavailability being thus secondary to this effect on ROS. Consequently, the reduction in Opa1 level in mice led to a decrease in FMD. In agreement, we found that ROS reduction with SOD and catalase restored FMD in EC-Opa1 mice and that counteracting the effect of ROS with L-Arginine and BH4 had a similar effect. These observations further support that Opa1 might be involved in flow mediated reduction in ROS production rather that in the direct activation of the NO pathway by flow. Finally, we also found that MitoTempo restored FMD to control levels in arteries isolated from EC-Opa1 mice, thus suggesting that flow reduced the production of ROS by the mitochondria. This agrees with our previous work showing that flow-mediated reduction in ROS involves the non-nuclear estrogen receptor alpha through a reduction in mitochondrial ROS production [33,34]. Nevertheless, the link between flow (shear stress), Opa1 and mitochondrial ROS production remains to be further investigated. Similarly, the interaction between this pathway and the non-nuclear estrogen receptor alpha remains unknown. Of note, we found no difference between male and female mice in the response to flow in mice lacking Opa1.

These findings led us to investigate the role of Opa1 in pathological conditions. Shear stress due to flow is the major regulator of ECs homeostasis. Nevertheless, flow is disturbed in zones of branching, bifurcations and curvatures, leading to atherosclerosis susceptibility [47]. Indeed, shear stress due to laminar or pulsatile flow induces a protective signal in ECs, whereas a low or oscillatory shear stress induces a deleterious signal [63]. We investigated Opa1 expression in mouse ECs in vitro and in vivo. Opa1 expression level was reduced in mouse ECs exposed in vitro to disturbed flow. Compared to laminar flow, the disturbed flow was characterized by reduced eNOS and increased ET-1 expression level. In agreement, Opa1 expression was decreased in the lower curvature of the mouse aortic cross which is an atheroprone zone compared to the greater curvature which is less susceptible to atherosclerosis. This was associated with reduced mitochondrial length and increased fission count in ECs of the mouse aortic cross.

Consequently, disturbed shear stress is likely to reduce Opa1 levels leading to increased oxidative stress, thus decreasing the protective effect of shear stress on ECs. This was confirmed by our results showing that both the mouse model of ADOA (*Opa1*^+/−^ mice) and mice lacking Opa1 in ECs were more susceptible to atherosclerosis when fed a high-fat diet. Interestingly we observed a progressive protective effect of Opa1 along the aorta with a maximal protection in the descending aorta. We observed large areas of lipid deposits in the aortic sinus without effect of the absence of Opa1 (*Opa1*^+/−^ mice) whereas lipid deposition in the aortic cross was 36% greater in *Opa1*^+/−^ mice compared to *Opa1*^+/+^ mice. Finally, in the descending aorta the lipid deposition was increased by 113% in the absence of Opa1. A similar pattern was observed in EC-Opa1 mice. Thus, it is most likely that the protective effect of Opa1 was more pronounced in areas of the aorta with a more laminar flow as we also found that the expression level of Opa1 was reduced in ECs submitted to disturbed flow and increased in ECs submitted to laminar flow.

In conclusion, we found that Opa1 facilitates the acute response of arteries to flow (FMD) through a reduction in ECs oxidative stress. This finding suggests that a defect in FMD due to reduced flow-signaling of ECs might affect ADOA patients. This reduced FMD could explain, at least in part, the reduced retinal vascular density observed in ADOA patients [23,24]. These findings suggest that targeting Opa1 or more generally mitochondrial dynamics could provide new therapeutic tools against endothelial disorders which are involved in a handful of cardiovascular and cerebrovascular diseases.

## Figures and Tables

**Figure 1 antioxidants-11-01078-f001:**
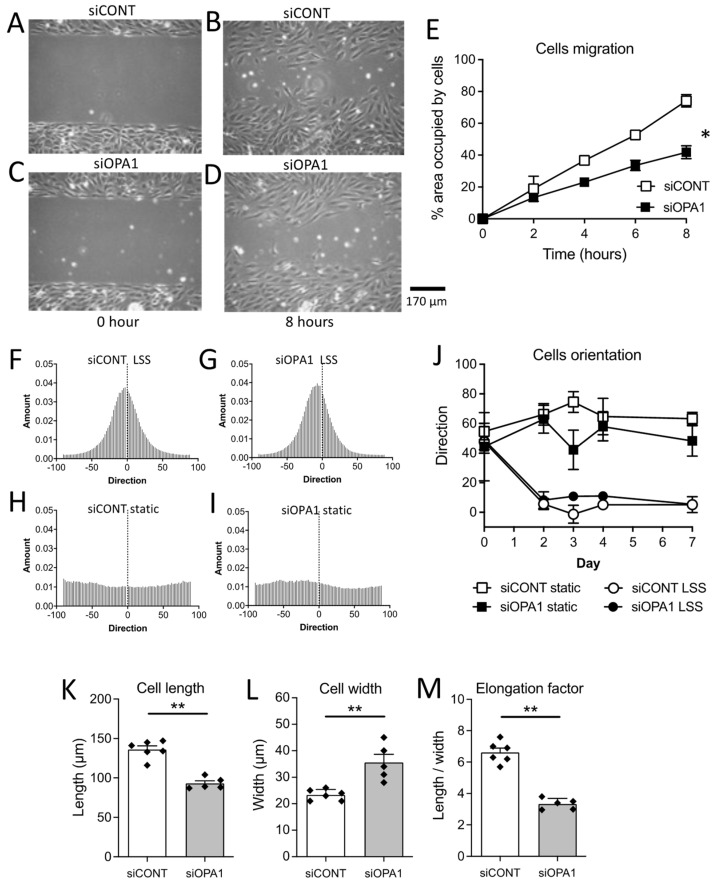
EC migration and flow-mediated EC alignment and elongation. Endothelial cells (HUVECs) were transfected for 72 h with the control scrambled siRNA (siCONT, **A**,**B**) or siOPA1 (**C**,**D**) and seeded in Ibidi^®^ μ-slides attachment (Ibidi, Munich, Germany). Cells were allowed to migrate, and imaged at different time points. Data are expressed as the area occupied by cells after each time point (**E**). Means ± SEM are shown (n = 3 independent experiments per group). Endothelial cell alignment (**F**–**J**) was determined after exposure to laminar flow (laminar shear stress, LSS, 20 dyn/cm^2^, 3 days, **F**,**G**), compared to static conditions (**H**,**I**). Quantification of cells’ orientation are shown in panel J. The cell elongation factor (**M**) was determined as the cell length (**K**) along flow direction divided by cell width (**L**). Means ± SEM are shown (n = 5 siOPA1 or 6 siCONT independent experiments per group for alignment and elongation measurements). Two-way ANOVA for repeated measurements (**E**,**J**): Panel **E**: * *p* = 0.0214 (interaction: *p* = 0.0001). Panel **J**: cell alignment (static versus flow in siCONT and siOpa1 cells). Panel **J**: NS, siCONT static versus siOpa1 static and siCONT flow versus siOpa1 flow. ** *p* = 0.0043, two-tailed Mann–Whitney test, panels **K**, **L** and **M**.

**Figure 2 antioxidants-11-01078-f002:**
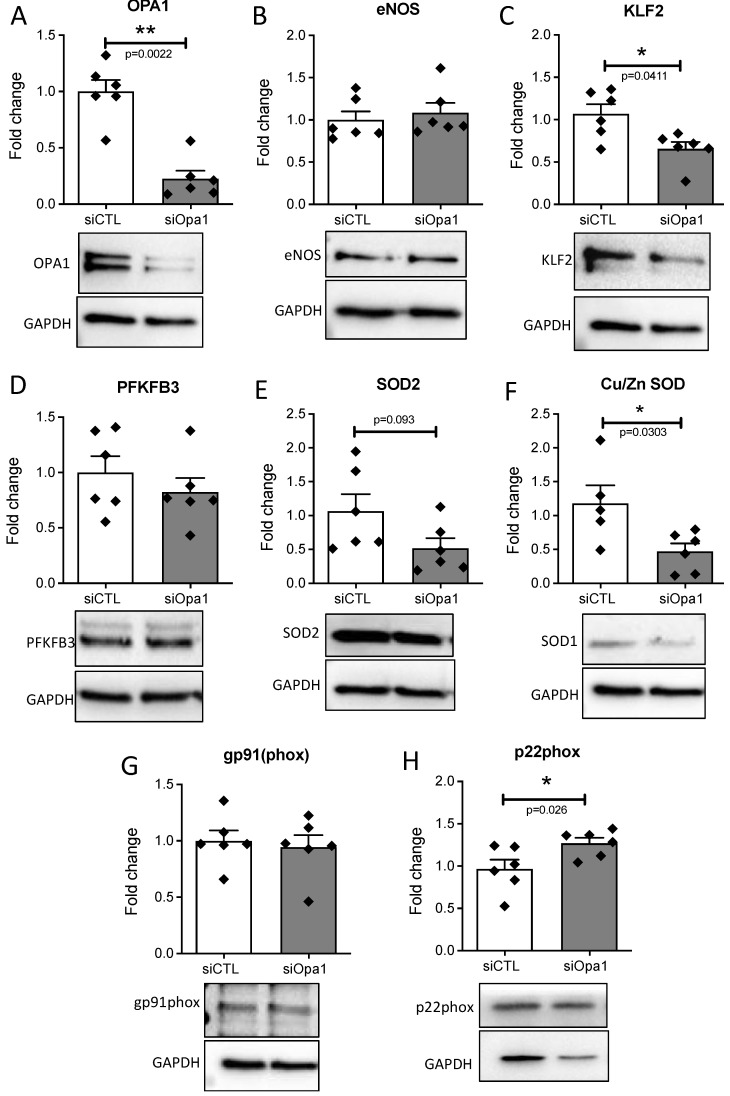
Protein expression measurements in endothelial cells submitted to laminar flow. (**A**–**H**) Protein expression levels of OPA1, eNOS, KLF2, PFKFB3, SOD2, SOD1, gp91phox, and p22phox were determined in HUVECs submitted to laminar flow using bidi µ-slides for 24h after OPA1 silencing (siOPA1) compared to the control scrambled siRNA (siCONT). Means ± SEM are shown (*n* = 6 independent experiments per group). Whole blots are shown in Appendix A. Two-tailed Mann–Whitney tests, * *p* > 0.05, ** *p* < 0.01 (*p*-values are shown on the graphs when *p* < 0.05).

**Figure 3 antioxidants-11-01078-f003:**
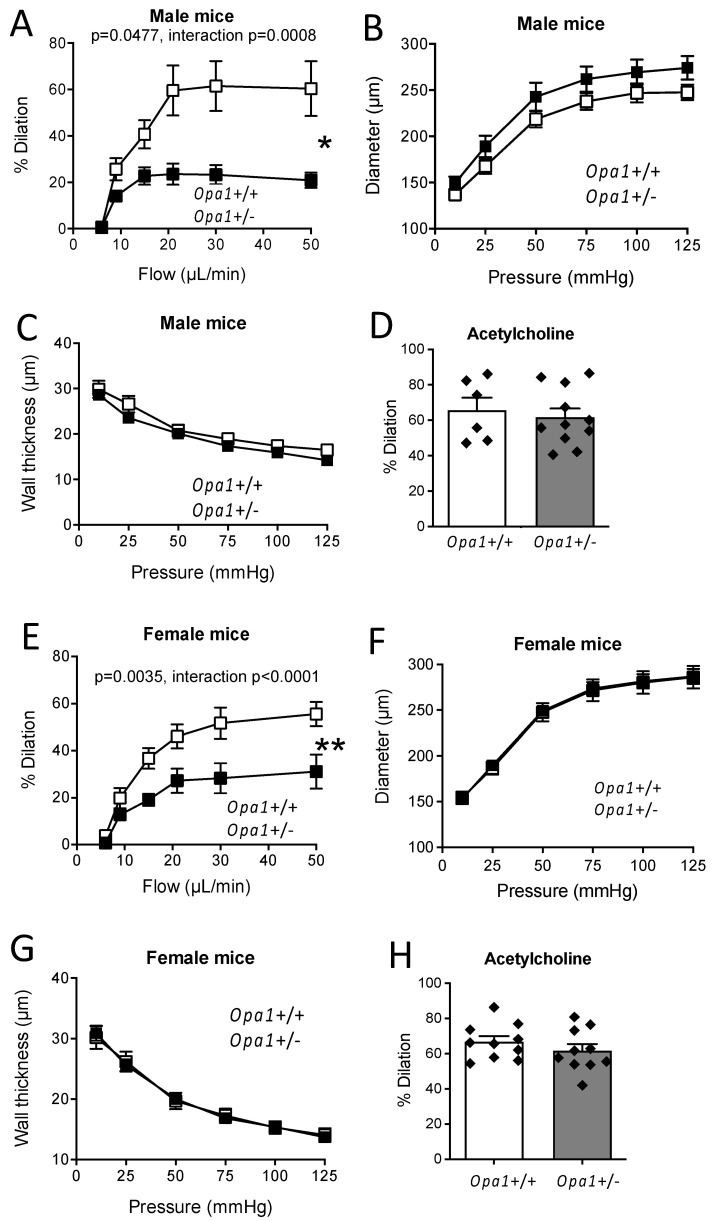
Effect of Opa1 haploinsufficiency on flow-mediated dilation. Flow-mediated dilation was measured in mesenteric resistance arteries isolated from *Opa1*^+/−^ and *Opa1*^+/+^ male (**A**) and female (**E**) mice. In the same arteries, arterial inner diameter (**B**,**F**) and wall thickness (**C**,**G**) were measured in response to stepwise increases in pressure. (**D**,**H**): Acetylcholine (1 µM)-mediated dilation. Means ± SEM are shown (n = 6 male EC-WT, 11 male EC-Opa1, 10 female EC-WT and 10 EC-Opa1 mice). Two-way ANOVA for repeated measurements (**A**–**C**,**E**–**G**). NS, two-tailed Mann–Whitney test, (**D**,**H**).

**Figure 4 antioxidants-11-01078-f004:**
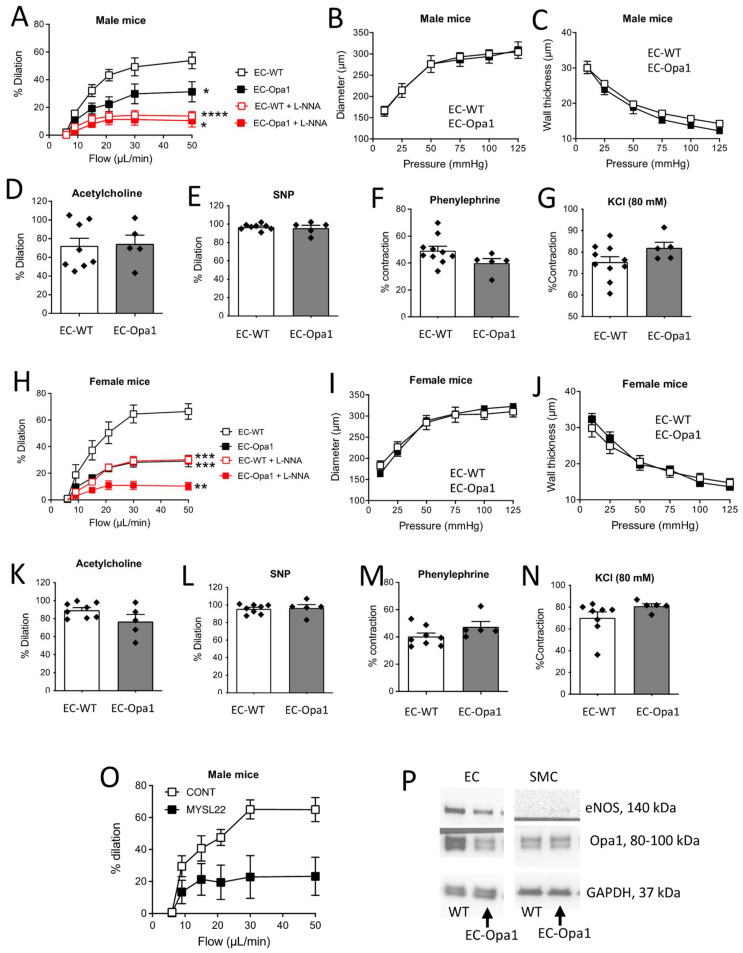
Consequence of Opa1 deficiency in endothelial cells on flow-mediated dilation. Vascular reactivity and structure were measured in mesenteric resistance arteries isolated from EC-WT and EC-Opa1 male (**A**–**G**,**O**) and female (**H**–**N**) mice. FMD (**A**,**H**) was determined in response to stepwise increases in flow in the presence or absence of the NO synthesis blocker L-NNA 100 µM, 30 min). In the same arteries, arterial inner diameter (**B**,**I**) and wall thickness (**C**,**J**) were measured in response to stepwise increases in pressure. **D**,**K**: Acetylcholine (1 µM)-mediated dilation. **E**,**L**: sodium nitroprusside (SNP, 10 µM)-mediated dilation. **F**,**M**: Phenylephrine (1 µM)-mediated contraction. **G**,**N**: KCl (80 mM)-mediated contraction. In a separate series of experiments, the effect of the Opa1 inhibitor MYSL22 (1 µM, 30 min) was tested on FMD (**O**). Panel **P**: Validation of Opa1 extinction in mesenteric resistance arteries endothelial cells (EC) from EC-Opa1 mice compared to smooth muscle cells (SMC) isolated from the same arteries. As a control for the identity of EC, eNOS protein expression was also measured together with GAPDH as a loading marker (**P**). Full blots are shown in Appendix A. Means ± SEM are shown (n = 8 male EC-WT, 5 male EC-Opa1, 8 female EC-WT and 5 EC-Opa1 mice in panels A to N and n = 4 male EC-WT with MYSL22 and 5 male EC-WT with solvent or control in panel **O**). Two-way ANOVA for repeated measurements: Panel A: * EC-Opa1 vs EC-WT: *p* = 0.0213, interaction *p* = 0.0045, **** LNNA in EC-WT: *p* < 0.0001, interaction *p* < 0.0001; * LNNA in EC-Opa1: *p* = 0.0198, interaction *p* = 0.0065. Panel H: *** EC-Opa1 vs EC-WT *p* = 0.00006, interaction *p* < 0.0001; *** LNNA in EC-WT: *p* = 0.0007, interaction *p* < 0.0001; ** LNNA in EC-Opa1: *p* = 0.0028, interaction *p* < 0.0001. Panel **O**: * *p* = 0.0221, interaction *p* = 0.0228. NS: Two-way ANOVA for repeated measurements (**B**,**C**,**I**,**J**). NS, two-tailed Mann–Whitney test, panels **D**–**G** and **K**–**N**.

**Figure 5 antioxidants-11-01078-f005:**
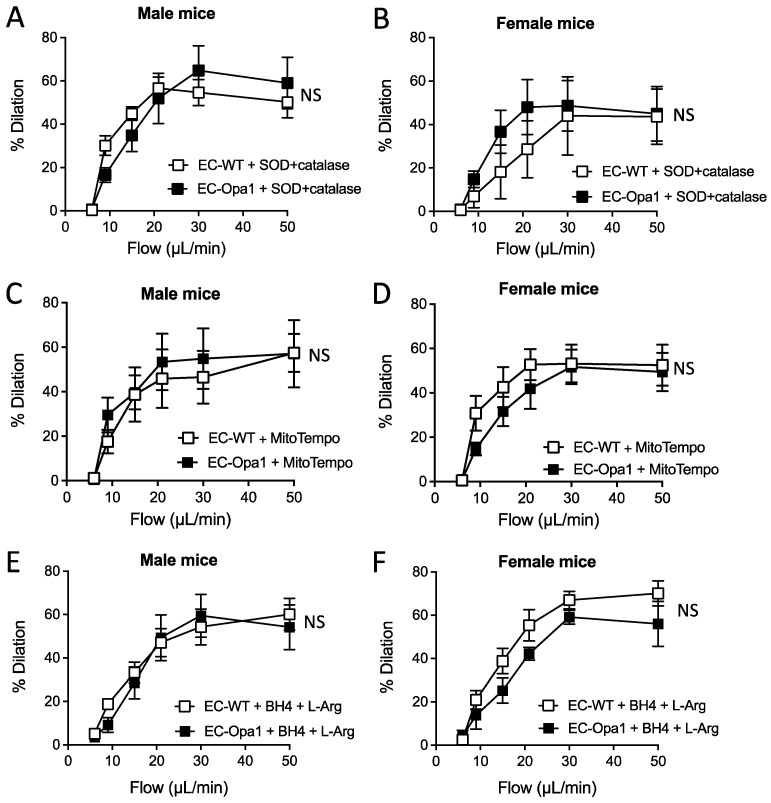
Consequence of reducing oxidative stress on flow-mediated dilation. Flow-mediated dilation was measured in mesenteric resistance arteries isolated from EC-WT and EC-Opa1 male and female mice after a treatment with SOD (120 U/mL, 30 min) plus catalase (80 U/mL, min) (**A**,**B**), MitoTempo (1 µmol/L, 30 min, 5 **C**,**D**), or tetrahydrobiopterin (BH4, 10 µmol/L, 30 min) plus L-arginine (L-Arg, 100 µmol/L, 30 min) (**E**,**F**). Means ± SEM are shown (n = 5 mice per group). NS: Two-way ANOVA for repeated measurements.

**Figure 6 antioxidants-11-01078-f006:**
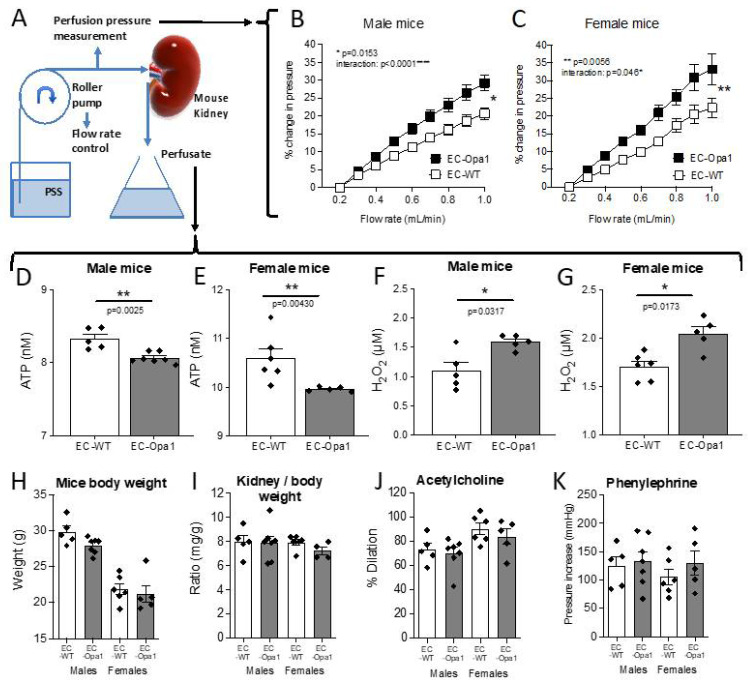
Consequences of Opa1 deficiency in endothelial cells on the kidney perfusion pressure and on kidney ATP and H_2_O_2_ production. In isolated and perfused kidneys (**A**), the flow-pressure relationship was determined in mice lacking Opa1 in endothelial cells (EC-Opa1 male, **B**, and female mice, **C**) and in littermate wild-type mice (EC-WT). The level of ATP (**D**,**E**) and H_2_O_2_ (**F**,**G**) was quantified in the perfusate collected from the perfused kidneys under a flow rate of 600 µL/min. **H**: mice body weight, **I**: ratio kidney/body weight, **J**: Acetylcholine (1 µM)-mediated dilation, and **K**: Phenylephrine (1 µM)-mediated contraction. Means ± the SEM are shown (n = 6 EC-WT and 7 EC-Opa1 male mice and n = 6 EC-WT and 5 EC-Opa1 female mice).* *p* >0.05, ** *p* < 0.01, two-way ANOVA for repeated measurements (panels **B**–**C**). * *p* > 0.05, ** *p* < 0.01, two-tailed Mann–Whitney test (panels **D**–**G**). NS: Kruskal–Wallis test, EC-WT versus EC-Opa1 (panels **H**–**K**).

**Figure 7 antioxidants-11-01078-f007:**
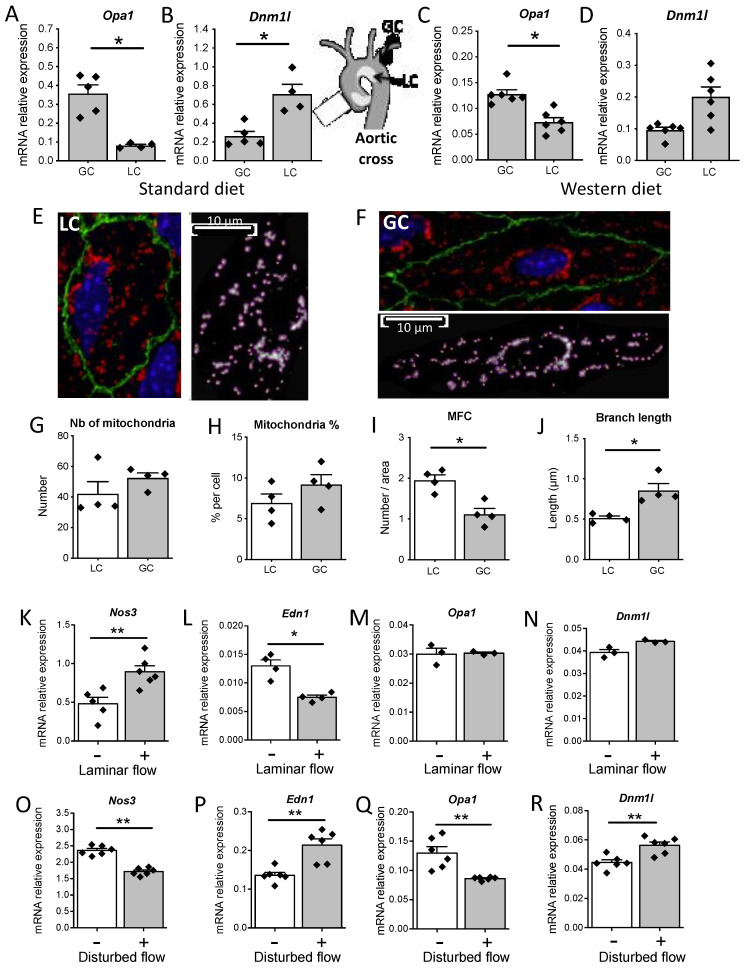
Disturbed flow and mitochondrial dynamics. Drp1 (*Dnm1l*) and Opa1 (*Opa1*) expression levels were determined in the lesser curvature (LC) and in the greater curvature (GC) of the mouse aortic cross. Mice were fed normal (**A**,**B**) or Western diet (**C**,**D**). In a separate series of experiments (**E**–**J**), mitochondrial shape was analyzed in the LC (**E**) and in the GC (**F**) and the number (Nb) of mitochondria (**G**), the % of mitochondria per cell (**H**), the mitochondria fission count (**I**) and the mean branch length (**J**) were determined. Finally, MS1 cells were submitted to laminar (**K**–**N**) or disturbed flow (**O**–**R**) using an orbital shaker and circular flow. After 72 h cells were collected for the measurement of eNOS (*Nos3*), endothelin-1 (*Edn1*), Opa1 (*Opa1*), and Drp1 (*Dnm1l*) expression levels. Means ± SEM are shown (**A**–**J**: n = 4 to 6 mice per group; **K**–**R**: n = 3 to 6 independent experiments). * *p* < 0.05 and ** *p* < 0.01, two-tailed Mann–Whitney test.

**Figure 8 antioxidants-11-01078-f008:**
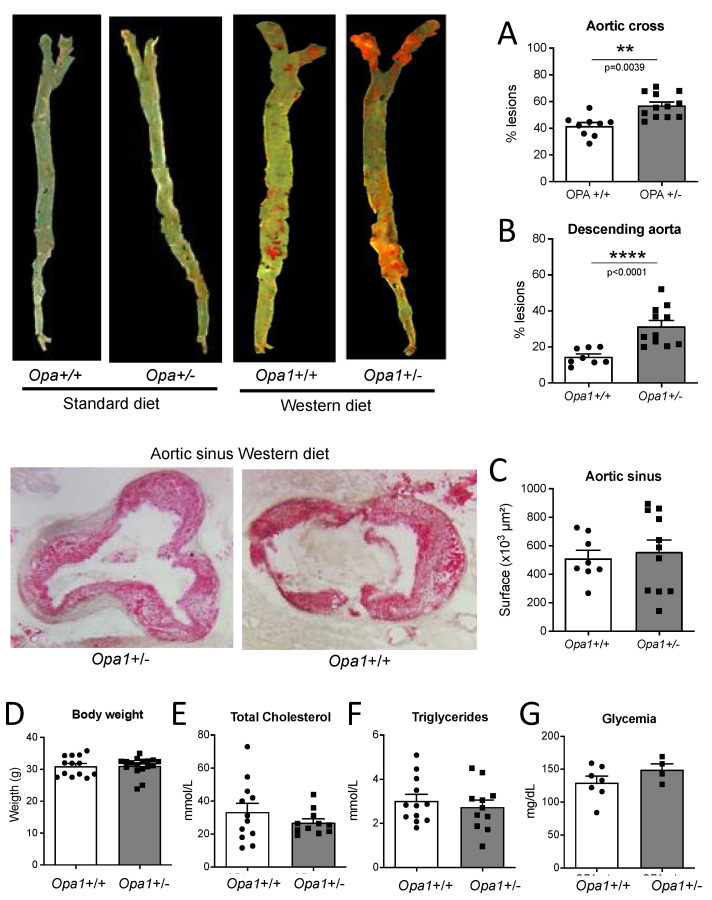
Consequence of Opa1 deficiency in endothelial cells on plaque formation. *Opa1^+^*^/*+*^
*and Opa1^+^*^/*−*^ mice that were three months old were fed with an atherogenic diet (Western diet) and compared to *Opa1*^+/+^
*and Opa1*^+/−^ mice fed with a standard diet. After 4 months the aorta was collected from the sinus to the iliac bifurcation and stained with Oil red-O. Lipid deposition was quantified in the aortic cross, (**A**), in the descending aorta (**B**) and in the aortic sinus (**C**). Bodyweight (**D**), plasma cholesterol (**E**), triglycerides (**F**), and glycemia (**G**) were measured in each group of mice. Means ± SEM are shown (n = 11 *Opa1*^+/−^ and 8 *Opa1*^+/+^ mice per group). ** *p* = 0.0039 (**A**); **** *p* < 0.0001 (**B**) and NS (**C**–**G**), Two-tailed Mann–Whitney test.

**Figure 9 antioxidants-11-01078-f009:**
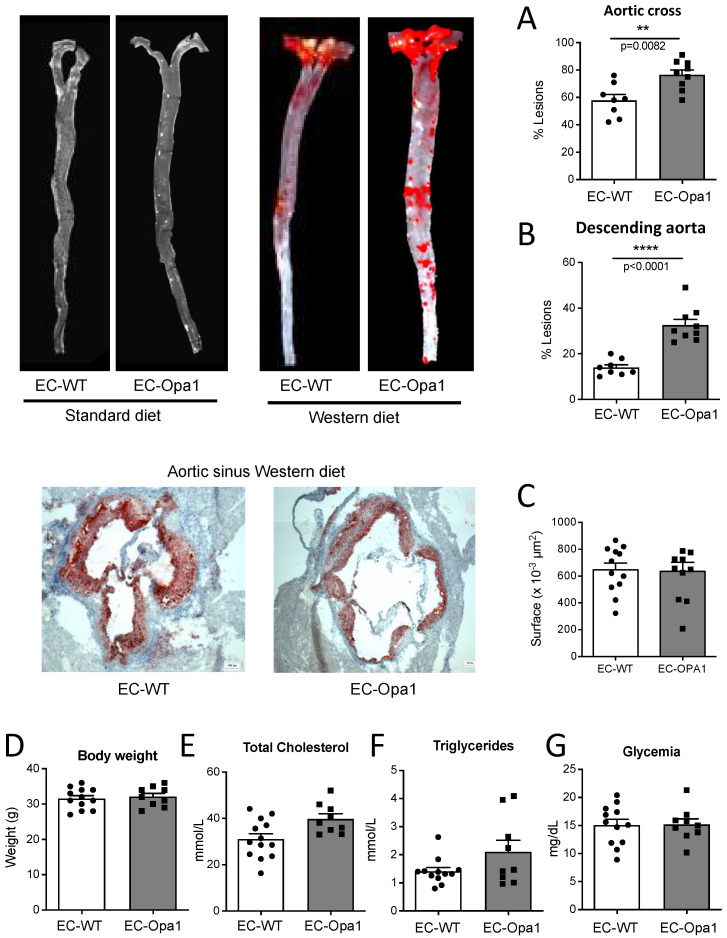
Consequence of Opa1 deficiency in endothelial cells on plaque formation. EC-Opa1 and EC-WT male mice that were three months old were fed with an atherogenic diet (Western diet) and compared EC-Opa1 and EC-WT mice fed with a standard diet. After 4 months the aorta was collected from the sinus to the iliac bifurcation and stained with Oil red-O. Lipid deposition was quantified in the aortic cross, (**A**), in the descending aorta (**B**) and in the aortic sinus (**C**). Bodyweight (**D**), plasma cholesterol (**E**), triglycerides (**F**), and glycemia (**G**) were measured in each group of mice. Means ± SEM are shown (9 EC-Opa1 and 8 EC-WT mice per group were used). ** *p* = 0.0082 (**A**); **** *p* < 0.0001 (**B**) and NS (**B**–**G**), Two-tailed Mann–Whitney test.

**Table 1 antioxidants-11-01078-t001:** List of the primers used for RT-qPCR.

Gene	*Accession number*	Forward	Reverse
*Nos3*	NM_008713.4	CCAGTGCCCTGCTTCATC	GCAGGGCAAGTTAGGATCAG
*Edn1*	NM_010104.4	TGCTGTTCGTGACTTTCCAA	GGGCTCTGCACTCCATTCT
*Opa1*	NM_001199177.1	ACCAGGAGAAGTAGACTGTGTCAA	TCTTCAAATAAACGCAGAGGTG
*Dnml1*	NM_001276340.1	AGATCGTCGTAGTGGGAACG	CCACTAGGCTTTCCAGCACT
*Hprt*	NM_013556.2	AAGACATTCTTTCCAGTTAAAGTTGAG	AAGACATTCTTTCCAGTTAAAGTTGAG
*Gapdh*	NM_008084.2	CCGGGGCTGGCATTGCTCTC	GGGGTGGGTGGTCCAGGGTT
*Gusb*	NM_010368.1	CTCTGGTGGCCTTACCTGAT	CAGTTGTTGTCACCTTCACCTC

**Table 2 antioxidants-11-01078-t002:** List of the antibodies used for the Western-blot analysis.

Target Antigen	Vendor or Source	Catalog #	Working Concentration	Lot #	Persistent ID/URL
eNOS	BD Biosciences	#610297	1/1000	8199630	AB_397691
OPA1	BD Biosciences	#612606	1/1000	1025917	AB_399888
Cyp1B1	Santa Cruz biotechnologies	#sc-374228	1/500	K0620	AB_10990317
PFKFB3	Cell Signaling Technology	#13123	1/1000	2	AB_2617178
Klf2	Atlas antibodies	#HPA055964	0.5 µg/mL		AB_2682989
SOD2	Cell Signaling Technology	#13141	1/1000	2	AB_2636921
SOD3	Enzo	#ADI-SOD-101-E	0.5 µg/mL	01101319	AB_2039584
gp91phox	BD Biosciences	#611414	1/500	6226646	AB_398936
gp91phox	AssayGenie	CAB11966	1/500	1	AB_2915942
p22phox	Cell Signaling Technology	#27297	1/1000	1	
GAPDH	Cell Signaling Technology	#2118	1/2000	14	AB_561053

#: Number.

## Data Availability

Data is contained within the article and Appendix A.

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
