# Peer review of "Altered Mitochondrial Opa1-Related Fusion in Mouse Promotes Endothelial Cell Dysfunction and Atherosclerosis"

_antioxidants, 2022, doi:10.3390/antiox11061078_

Round 1
Reviewer 1 Report
The authors have suitably responded to most of my comments, and as a result the manuscript is much improved.
There are however some remaining concerns.
1. While the authors have improved the detail on the N-numbers in the figure legend, it is still not apparently clear why the groups are not matched for number. For example, in Figure 3 the N-numbers span from 6-11. The reason for these differences in numbers needs to be explained clearly and justified.
2. In performing the orbital rotation, I would ask the authors whether they considered the parameters that influence forces such as shear stress when assembling their study? Well diameter, media viscosity, radius of rotation, etc? Subtle differences can lead to drastically different results in terms of physiological flow in this setup, so for clarity the shear stress is the more accurate measurement to use rather than RPM.
Author Response
Thank you for your comments and advices. We modified the manuscript accordingly as described below:
- While the authors have improved the detail on the N-numbers in the figure legend, it is still not apparently clear why the groups are not matched for number. For example, in Figure 3 the N-numbers span from 6-11. The reason for these differences in numbers needs to be explained clearly and justified.
Response: the number of mice per group is difficult to match when we perform arteriography on living blood vessels or with the perfused kidney experiments. Based on our previous experiments performed using these models in the past decades we started with a minimum of 10 to 12 mice per group for the arteriography and 8 for the kidney experiments (number of KO and WT in a litter is not always following the Mendel’s law) and we have a limited number of mice for breading as defined by the authorization obtained from the ethical committee. The next step is to sacrifice a mouse and to isolate either a kidney or a piece of mesentery.
For mesenteric arteries only 2-3 arterial segments were collected per mouse as the arterial viability decreases rapidly after the death of the mouse. Then, the mounting of an arterial segment on the glass cannula and the tests of the arterial viability are crucial before starting the experiments (FMD with or without pharmacological agent). The experiment can easily fail for the following reasons (here are listed the most common reasons for stopping the experiment):
- One glass cannula breaks during the mounting of the artery or after a washout (glass is only a few µm thick).
- pressure is not maintained due to a leak (small collateral arteriole not visible when mounting the artery).
- The artery does not pass a test: no contractility (smooth muscle damaged during dissection or mounting) or no dilation (endothelium damaged during dissection or mounting)
- Disconnection of the artery from one glass cannula during the test or after a washout of the PSS (PSS is changed every 20 min).
- Formation of gas bubbles in the tubing (the PSS is bubbled with air and CO2) and one bubble crosses the artery. This is enough to damage the endothelium and to stop the experiment.
After stopping the experiment another artery can be mounted if it’s not too late. As only a few arterial segments can be taken from one mouse, it happens that no experiment could be completed with one mouse. Therefore, the number of animals varies from group to group. This is very common with “physiological setups” using living tissues.
A paragraph was added to the text lines 125 to 136.
The same limitations apply to the perfused kidney. It’s a living tissue perfused in an organ bath, and it remains very fragile. Some of the problems listed above can apply to the kidney (gas bubble moving from the tubing to the kidney, no contractility or no dilation if the dissection or cannulation is too long, disconnection of the kidney from the cannula). Only one kidney can be mounted per mouse (blood clots in the 2nd one during the cannulation of the first one; cannulation is done in vivo under anesthesia). Consequently, the number of successful kidneys also varies in the different group.
A paragraph was added to the text lines 147 to 152.
Finally, with mice submitted to high fat diet the number of mice per group varies because of the litter size, and the proportion of mice with the good genotype (double knockout with Ldlr-/- and Opa1+/- or EC-Opa1) per litter also induces some variability in the number of mice available. In addition, 3 mice died before the end of the 4 months of feeding with the high fat diet (1 EC-WT-ldlr-/- and 2 Opa1+/+-Ldlr-/-).
- In performing the orbital rotation, I would ask the authors whether they considered the parameters that influence forces such as shear stress when assembling their study? Well diameter, media viscosity, radius of rotation, etc? Subtle differences can lead to drastically different results in terms of physiological flow in this setup, so for clarity the shear stress is the more accurate measurement to use rather than RPM.
Response: The model is detailed in reference 36. Using this reference, we calculated the shear stress in the conditions of laminar or protective flow (210 rpm corresponding to 12 dyn/cm2). With 260 rpm, the estimated shear stress is 18 dyn/cm2 but the flow is very disturbed along the periphery and close to zero in the center. Therefore, with 260 rpm we found an overexpression of endothelin-1 and a down regulation of eNOS. We added the shear stress value for 210 rpm in line 218. We did not add the shear stress value for 260 rpm as flow is too disturbed.
Reviewer 2 Report
The authors have addressed all major concerns with addition of significant new data that strengthens the outcomes of the original submission and strongly support the conclusions now drawn.
I find only one minor issue that requires attention.
1) Please add an appropriate "scale bar" to the cell images in Figure 1.
Author Response
We added the scale bar to figure 1.
Thank you for checking.
best regards
This manuscript is a resubmission of an earlier submission. The following is a list of the peer review reports and author responses from that submission.
Round 1
Reviewer 1 Report
The authors have presented a very in-depth examination of the role of Opa1; a key protein in mitochondrial dynamics, in the endothelial response to hemodynamic forces. Specifically, the authors examine a putative link between key proteins involved in dilation of blood vessels and Opa1 expression/activity using a variety of cell lines, flow models, and animal tissues, with Opa1 knockouts employed throughout. Ultimately, the authors found Opa1-driven fusion of mitochondria altered the dilatory response of the endothelium to flow, with an increased incidence of reactive oxygen species evident. Together, this suggests that Opa1 may represent a viable therapeutic target in cardiovascular-related diseases.
In reviewing the manuscript, I noted a number of concerns. The authors should address the following when preparing a suitable revision.
- The manuscript is quite detailed and well written for the most part, however, there are a number of grammatical errors within. The authors should review the manuscript and attempt to reduce the incidence of these.
- The authors employed a number of knockout animals models throughout the study – how was the absence of the protein determined? Specifically in the tissue of interest?
- It is unclear in the animal studies as to how many animals were used at times. I appreciate the inclusion of dot plots to validate this in some instances, however, there are several instances where it is not entirely clear, and moreover, in some a range is given for n-number. The authors must clarify the exact n-number in each study (in the figure legends for example), while justification for the difference in n-numbers within a study should also be provided.
- More details on the transfection protocol would be welcomed. For example, what concentration of siRNA was used in each experiment to achieve the levels of knockdown reported here. Also what control were used? Any scrambled or transfection controls performed to validate the model?
- Why were so many different cell type/lines used in this study?
- The authors employ orbital rotation as a method of shear application and mention a speed of 210-260rom was used – why the range? The authors should clarify how they calculated the rate of shear based on these speeds, the media viscosity, and other variables and explain why a range was utilised and not a fixed speed.
- In examining the Western blots in the supplementary data there is a lot of noise. The authors have used molecular weight it appears as a guide to determine which bands may correspond to their target protein, however given the degree of noise I would wonder if the bands are the target at all. Did the authors perform any experiments where a recombinant form of the protein for example was used to validate the antibodies being employed.
- More details on the measurement of the mitochondria would be welcomed. The authors use a reference but the method is quite light. It would be preferable is even a brief description were given such that readers had access to the method without requiring to potentially purchase/access another research paper.
- Was there any particular reason that the Y-slides of the ibidi were utilised in this study? The y-design will generate oscillatory flow while the focus of this study appeared to be examining the effect of flow in general.
- Were the primers utilised in this study examined and verified to be MIQE guidelines compliant?
- It would be useful if in the results section it was clarified in the flow-mediated experiments whether the ibidi or orbital rotation method of shear stress was applied.
- While the depth of the data set is quite good, some of the figure panels are difficult to read/analyse given the scale of some of the graphs to accommodate so much. It might be useful to split some data across more than one panel, or else include some as supplementary to address this.
Reviewer 2 Report
This is an excellent collection of data from an active research group. In general the data has been carefully compiled and the use of both cellular and animal models complement each other to yield an interdisciplinary research manuscript is commended. Several issues have been identified as follows:
- Intro - line 73 minor issue: perhaps "characteristics" should read - phenotypic characteristics in humans.
- Given some mouse (x)breeding was conducted here - Was genotyping of offspring conducted to confirm the assignment?
- There are several instances in the results / data set presentation where a measure of NO bioactivity would complement the data presented - Perhaps a measure of cGMP here would be useful to confirm that NO bioavailability is compromised.
- Was a scrambled RNAi control employed? Identifying mitochondrial dysfunction using a probe that ascertains membrane potential would also complement the assessments of endothelial cell dysfunction.
- Cell migration assay - It would be good to have a negative/positive control to show an anticipated inhibited/enhanced cell growth under the conditions employed here.
- Assessment of gene regulation - Better to add a Table and tabulate the forward and reverse primers and corresponding accession numbers from the Blast searches.
- A general comments on lack of units - % is % w/v, dilutions are usually v/v unless stated as g/mL. Also there is inconsistent use of min and minutes - possibly hours and h as well. Pick one format and stick to it.
- Vascular lesions: It would be appropriate to also assess lesion development histologically as this will provide information on the composition of the lesions and not just areal fraction. Composition can be related to lesion stability which is important in determining the likelihood of rupture and acute thrombotic events when extrapolating to the human condition.
- Section 3.3:
These data are consistent with an increase in superoxide radical anion interfering with FMD - any evidence that cGMP is decreased and NO bioactivity is being altered - eg., through the production of ONOO- (itself a potent oxidant that can nitrate proteins and oxidise redox sensitive thiol residues).
Does the eNOS data support monomerisation thereby changing the active to inactive form? The dimer is detected at ~260 kDa, which seems to be absent in the blots shown in the main and supplementary data sets.
- Why no positive control here with the eNOS inhibitor? How do the FMD and ACh data mesh together? If ROS increases explains the change in FMD why then is Ach stimulation able to show NS different dilation? What about endothelium independent relaxation ?
- Fig 4 - Is ENOS the monomer here ? The dimeric form is the active NO producing enzyme at ~266 kDa - corresponding supplemental data shows no active eNOS dimer. This will limit the utility of the data to make strong conclusions wrt eNOS activity/NO production and activity - this is where measures of cGMP may be useful as a biomarker for NO activation of soluble guanylyl cyclase.
- NO discussion on the role for superoxide radical anion to alter NO bioavailability - bioactivity as per my comments above.